# Cell-Type-Specific Profibrotic Scores across Multi-Organ Systems Predict Cancer Prognosis

**DOI:** 10.3390/cancers13236024

**Published:** 2021-11-30

**Authors:** Huihui Fan, Peilin Jia, Zhongming Zhao

**Affiliations:** 1Center for Precision Health, School of Biomedical Informatics, The University of Texas Health Science Center at Houston, Houston, TX 77030, USA; Huihui.Fan@uth.tmc.edu (H.F.); peilin.jia@gmail.com (P.J.); 2Human Genetics Center, School of Public Health, The University of Texas Health Science Center at Houston, Houston, TX 77030, USA; 3MD Anderson Cancer Center, University of Texas Health Graduate School of Biomedical Sciences, Houston, TX 77030, USA

**Keywords:** profibrotic cellular phenotype, single-cell extracellular matrix pathway activity, fibrosis, pan-cancer, cancer prognosis

## Abstract

**Simple Summary:**

Fibrosis is a major player and contributor in the tumor microenvironment. Profibrotic changes precede the early development and establishment of a variety of human diseases, such as fibrosis and cancer. Being able to measure such early signals at the single cell level is critically useful for identifying new mechanisms and potential drug targets for a wide range of diseases. This study was designed to computationally identify profibrotic cell populations using single-cell transcriptomic data and to identify gene signatures that could predict cancer prognosis.

**Abstract:**

Fibrosis is a major cause of mortality. Key profibrotic mechanisms are common pathways involved in tumorigenesis. Characterizing the profibrotic phenotype will help reveal the underlying mechanisms of early development and progression of a variety of human diseases, such as fibrosis and cancer. Fibroblasts have been center stage in response to various stimuli, such as viral infections. However, a comprehensive catalog of cell types involved in this process is currently lacking. Here, we deployed single-cell transcriptomic data across multi-organ systems (i.e., heart, kidney, liver, and lung) to identify novel profibrotic cell populations based on ECM pathway activity at single-cell resolution. In addition to fibroblasts, we also reported that epithelial, endothelial, myeloid, natural killer T, and secretory cells, as well as proximal convoluted tubule cells of the nephron, were significantly actively involved. Cell-type-specific gene signatures were enriched in viral infection pathways, enhanced glycolysis, and carcinogenesis, among others; they were validated using independent datasets in this study. By projecting the signatures into bulk TCGA tumor samples, we could predict prognosis in the patients using profibrotic scores. Our profibrotic cellular phenotype is useful for identifying new mechanisms and potential drug targets at the cell-type level for a wide range of diseases involved in ECM pathway activation.

## 1. Introduction

Fibrogenic responses can be triggered by disease-related injury in any organ. When injury progresses over a prolonged period, inevitable scarring occurs; thus, it causes cellular dysfunction and organ failure [1]. With approximately 45% of all-cause human mortality attributed to fibrosis [2], no effective anti-fibrotic therapies, especially in the lung [3], are currently available to limit the fibrogenesis in response to tissue injury. Meanwhile, chronic fibrosis serves as a major component of the tumor microenvironment (TME), which further impacts and promotes tumorigenesis [4,5]. Therefore, an extensive understanding of the prolonged fibrogenesis process to identify potential cell types and cell-type-level mechanisms involved in promoting fibrosis, especially in its early stages, is urgently needed.

A plethora of diseases in different organ systems are associated with fibrotic changes. These include, but are not limited to, the commonly reported the tissue of the lung [6], liver [7], kidney [8], and heart [9]. During the pathological process of chronic tissue fibrogenesis, immediate damage to the epithelial/endothelial barrier represents one of the early events accompanied by releasing the fibrogenic cytokine transforming growth factor beta (TGF-β1) [10]. Tissue damage could be induced by multiple factors, among which viral infections, as well as autoimmune and inflammatory stimuli, can be counted as common causes [11]. Various cell types in different organs synthesize TGF-β1, which plays a critical role in orchestrating the cross-talk between parenchymal, proinflammatory, and profibrotic cells [12]. Given the central role of fibroblast cells in depositing extracellular matrix proteins, its activation and proliferation in fibrosis has been studied [13,14,15]. Importantly, thanks to the massive production of high-resolution single-cell transcriptomic sequencing data (scRNA-seq), especially during the COVID-19 pandemic, more granular mechanism studies in fibrosis, down to the cell-type level, are gaining power. These data now are critically useful in revealing the complex cell network that underpins fibrosis in multiple organ systems [16,17,18]. In addition to fibroblasts and resident myofibroblasts, a diversity of cell types, such as epithelial and mesenchymal cells in the lung [19] and endothelial and tubular cells in the kidney [20], have been reported to contribute to pathologic fibrosis.

In this study, we repurposed and analyzed approximately 150,000 single-cell transcriptomes across organs of the lung, liver, kidney, and heart in response to the severe and acute COVID-19 viral infection. This investigation helped build a comprehensive understanding of the diversity of cell types exhibiting profibrotic phenotype in response to external stimuli. By adapting single-pathway enrichment analysis in single samples to the single-cell level, we computationally defined profibrotic cells contributing to the pathological fibrogenesis as active in expressing ECM pathway genes. A panel of seven different cell types was then identified to be involved in the process: fibroblast, epithelial, endothelial, myeloid, natural killer T (NKT), and secretory cells, as well as proximal convoluted tubule cells of the nephron (PCT-S3). Molecular pathway enrichment analysis based on cell-type-specific differentially expressed genes and co-active gene set analysis emphasized the functional relevance of the profibrotic cellular phenotype to fibrosis and human cancer. Validation using independent scRNA-seq datasets confirmed that our gene signatures could capture early profibrotic changes using both the human and mouse fibrosis data. We then projected cell-type-specific differential gene signatures into bulk tumor samples. This process helped us predict prognosis in tumor patients from The Cancer Genome Atlas (TCGA) consortia. Using high-resolution transcriptomic data at the single-cell level and the identification of profibrotic cell populations provided us with some novel insights into the cellular architecture and the fundamental mechanisms of the early profibrotic changes across multiple organs in response to severe viral infection.

## 2. Materials and Methods

### 2.1. Multi-Organ Tissue Atlas at Single-Cell Resolution

CellRanger-mapped [21] and CellBender background-removed [22] count matrices were downloaded from the Gene Expression Omnibus (GEO, https://www.ncbi.nlm.nih.gov/geo/; accessed 25 May 2021) under the accession number GSE171668. This multi-organ tissue atlas included publicly available processed single-cell or single-nucleus RNA sequencing (scRNA-seq or snRNA-seq) data of lung (*n* = 16 donors, k = 106,792 cells/nuclei), heart (*n* = 18, k = 40,880), liver (*n* = 15, k = 47,001), and kidney (*n* = 16, k = 33,872) [23]. Manual annotations of cell types were inherited from the original study as well.

All patients who succumbed to SARS-CoV-2 infection were confirmed by the qRT-PCR assays. A cross-body post mortem autopsy collection within 24 h, spanning 11 organs and 17 donors, was used to generate this single-cell tissue atlas.

### 2.2. Preprocessing the Single-Cell Tissue Atlas Data

We further filtered out cells/nuclei with fewer than 400 UMIs or 200 genes, or greater than 20% of UMIs mapped to mitochondrial genes. scRNA-seq or snRNA-seq data from individual samples were then combined and batch-corrected using the R package Harmony (v0.1.0) [24]. The harmonized single gene expression matrix was normalized against library size and corrected for cell cycle phase scores (using the CellCycleScoring function in R package Seurat v4.0.3 [25]) to remove the unwanted cycling bias among proliferating cells.

### 2.3. Identifying Profibrotic Cell Populations and Their Transcriptomic Signatures

Cell populations actively expressing ECM genes are defined as ECM-active cells, or alternatively, profibrotic cells. To identify these unique cell populations within each cell type, we adapted the concept of single-sample single-pathway activity analysis to the single-cell level. Pooled single cells across multi-organ systems were clustered based on the top 50 principal components, which resulted in a total of 43 clusters. The ECM pathway gene set was downloaded from the MSigDB (v7.4) c2.cp collection [26], which included 170 genes in total. The pathway enrichment score for the ECM pathway in each cell was then calculated using the Mann-Whitney-Wilcoxon Gene Set test (MWW-GST), implemented in the R package yaGST (v2017.08.25). MWW-GST requires both a gene set and a ranked gene list as inputs. When adapting the approach to the single-cell level, the expression of each gene was standardized within each of the 43 cell clusters to generate a Z score that was used to determine a single-cell gene rank. The resulting normalized enrichment score (*NES*) was an estimate of the probability that the expression of genes in the ECM pathway was greater than the expression of genes outside the ECM pathway:$$NES=1-\frac{U}{mn}$$
while
$$U=nm+\frac{m\left(m+1\right)}{2}-T$$
where *m* is the number of genes in the ECM pathway and *n* is the number of those outside the gene pathway. *T* is the sum of the ranks of genes in the ECM pathway. As the measure of pathway activity, NES ranges from 0 to 1, with values near 0 indicating minimal activation of the pathway and values near 1 meaning maximum activation of the pathway. Since single-cell matrices are extremely sparse, we defined a pathway with an NES score of larger than 0.3 and an adjusted *p* value less than 0.05 as extensively activated.

In addition to the ECM gene pathway, we downloaded the whole collection of pathways within the c2.cp category from the MSigDB (v7.4), which included 2064 pathways with more than 15 genes in total. The same procedure of calculating NES score to measure the pathway activities in every single cell was then performed accordingly.

Differentially expressed genes (DEGs) between ECM-active and non-ECM-active cell populations within each cell type were calculated, with an FDR cutoff of 0.05. Specifically, for cell type PCT-S3-1 in the kidney, we compared PCT-S3-1 to PCT-S3-2 to identify DEGs. The PCT-S3-1 was identified as ECM-active, while the PCT-S3-2 was treated as its counterpart within the same cell type, which represents an ECM-inactive cell population with the proinflammatory feature [27].

### 2.4. Constructing Cell-Type-Specific Profibrotic Co-Active Pathway Network

The activity of pathways downloaded from the MSigDB database was scored at the single-cell level individually, using the same processing pipeline as the ECM pathway. With a NES score higher than 0.3 and an FDR cutoff of lower than 0.05 (see above), we identified molecular pathways activated in cell types with ECM pathway genes actively expressed. A Fisher’s exact test was then carried out to test whether the association between the activity of the ECM pathway and that of a particular molecular pathway was statistically significant. The threshold for FDR-corrected *p* values was set as 0.05. The percentage of individual pathways being active against the total number of ECM-active cells in a certain cell type was then calculated to evaluate the prevalence of their associations. Top-ranked active pathways, based on both their significance and proportions, were then used to construct a cell-type-specific co-active pathway network. Only the top 10% of pathways, with a maximum number of 10, were shown in the Circos plot. Links were shown if more than 80% of ECM-active cells were shared between the two pathways. The bandwidth of the links was scaled to the percentage of shared cells.

### 2.5. Validating Cell-Type-Specific Gene Signatures in Both Human and Mouse scRNA-Seq Data

To validate that our cell-type-specific gene signatures could capture profibrotic changes, we collected three additional publicly available scRNA-seq datasets involving lung fibrosis. Time-series scRNA-seq data of the bleomycin-induced lung fibrosis in mice were collected at seven different time points (day 0, 3, 7, 10, 14, 21, 28; day 0 indicated no treatment). This dataset contained 29,297 cells spanning 28 different cell types [28]. Human single-cell atlas of idiopathic pulmonary fibrosis (IPF) profiled 312,928 cells from 32 IPF, 28 controls, and 18 chronic obstructive pulmonary disease (COPD) lungs [29]. A separate collective set of 1781 human fibroblast cells across different disease courses related to lung fibrosis was extracted from a single-cell study involving patients with IPF (*n* = 12), chronic hypersensitivity pneumonitis (cHP; *n* = 3), nonspecific interstitial pneumonia (NSIP; *n* = 2), sarcoidosis (*n* = 2), unclassifiable chronic interstitial lung diseases (ILD; *n* = 1), and nonfibrotic controls (*n* = 10) [19]. Processed single-cell datasets from all three available sources were downloaded and applied in the validation process. For the mouse data analysis in particular, gene signatures were matched between the human and mouse using R package biomaRt (v2.46.3).

NES was then calculated for each individual cell, as outlined in Section 2.3, using cell-type-specific profibrotic gene signatures as input. The normalized data count matrix was scaled within each cell type before ranking genes in the individual cell. As a surrogate of gene signature activity, NES was compared and plotted across different time points in mice and across different disease courses involving lung fibrosis in human single-cell datasets. Pairwise comparisons using non-parametric statistical tests with the FDR correction was carried out using R package ggstatsplot (v0.9.0). Significance level was set as 0.05 with FDR correction.

### 2.6. Calculating Cell-Type-Specific Profibrotic Phenotype Scores in Bulk TCGA Tumors

We utilized CIBERSORTx [30] for the calculation of cell-type-specific profibrotic phenotype scores using the DEGs between ECM-active and non-ECM-active populations within the same cell type. The gene marker file was derived from the single-cell tissue atlas data, with each row representing a DEG and each column representing a cellular phenotype, either ECM-active or non-ECM-active, spanning seven profibrotic cell types identified in our study. A total of 2514 genes were included in the marker file. The mixture gene expression file, with 10,325 bulk tumor samples in total, which contained 31 TCGA cancer types with more than 50 patient samples, was also uploaded to the CIBERSORTx portal. CIBERSORTx was used with the following options: relative mode, 100 permutations, and quantile normalization disabled [31]. One hundred permutations were used as the recommended minimum in order to achieve the balance of both accuracy and time consumption.

### 2.7. Survival Analysis

Profibrotic phenotype score matrix was retrieved from the CIBERSORTx website. We split the individual cell-type-derived profibrotic score based on the patients’ 25th and 75th percentile values into two groups within each tumor type and then performed the Kaplan-Meier analysis using overall survival time. Overall survival (OS) was defined as the time between the date of surgery and date of death or the date of the last follow-up. *p* values less than 0.05 were considered statistically significantly different between the two groups.

All data analyses were performed in the R programming environment (Version 4.0.5 or higher) [32] and Bioconductor.

## 3. Results

### 3.1. Identification of Novel Profibrotic Cell Populations Induced by Severe Acute Viral Infection

By leveraging large-scale, single-cell transcriptomic data across multiple organs simultaneously from patients who succumbed to SARS-CoV-2 viral infection, we computationally defined cells that exhibit profibrotic phenotype as those that are significantly active in expressing extracellular matrix (ECM) genes. To overcome the imbalance of this large and integrated single-cell data set, we first removed batch effects as described in Materials and Methods. Then, we clustered all the single cells extracted from the tissue of heart (*n* = 31,273 cells), lung (*n* = 65,743 cells), liver (*n* = 32,035 cells), and kidney (*n* = 21,635 cells), based on the top 50 principal components (Figure 1A). A total of 43 cell clusters was yielded, without any noticeable batch effects towards certain tissue types (Figure 1B). For instance, Clusters 13 and 16 are unique to the liver, because they are two subtypes of hepatocytes. Cluster 2 is a fibroblast cluster, which is a collection of fibroblast cells from four different tissue types [23].

With the focus on the ECM pathway, we showcased the gene expression variance within the pooled dataset (Figure 1C) as well as within each organ system (Figure 1D–G). As indicated, the difference in the most variable ECM genes under different biological conditions suggests a usage preference by different organ systems. For example, *MUC5B* expression is highly variable in the liver, while *MUC16* is the top one in the heart. To adopt a more objective way to evaluate the activity of the whole ECM pathway at a single-cell level, we adapted the approach to calculate the single-pathway activity in single samples but at the single-cell level (SSA). To this end (Figure 1H), we first identified potential maximum cell clusters, as shown in Figure 1A, followed by a Z-score transformation of the gene expression within each cell cluster. We then ranked each cell based on its relative Z-scores of all the genes in a decreasing manner. Normalized enrichment score (NES) and significance level by the Mann-Whitney-Wilcoxon test were calculated at single-cell resolution (see Section 2 for details). The ECM-active cell population was thus defined as those cells with an NES score higher than 0.3 and a significance level less than 0.05 after multiple testing correction using FDR (Figure 1I). In total, we identified 1404 ECM-active cells (Figure 1J), which spanned 7 different cell types: epithelial (*n* = 374), endothelial (*n* = 699), myeloid (*n* = 115), secretory (*n* = 28), natural killer T (NKT; *n* = 85), and fibroblast (*n* = 17) cells in the lung, fibroblast (*n* = 22) cells with low unique molecular identifier (UMI) counts (median UMI around 600) in the heart, and PCT-S3-1 (proximal convoluted tubule cell of the nephron, segment 3, Cluster 1; *n* = 64) in the kidney (Figure 1K). Interestingly, the PCT-S3 cell population in the original study split into two clusters [23]. Cluster 2 of the PCT-S3 cell in the kidney expressed *CFH* with fibrinogens (FGB, FGA) positive, which indicates a proinflammatory polarization. On the contrary, we found that in Cluster 1, 95.5% (64/67) of the PCT-S3 cells in the kidney actively expressed ECM pathway genes. This result indicated a novel profibrotic phenotype.

Among those cell types, fibroblast cells in the heart and lung have been repeatedly demonstrated to contribute to the pathogenesis of fibrosis, and their potential origins and roles are also well studied [33,34]. ECM-producing epithelial cells have been reported to be highly enriched in the lungs of pulmonary fibrosis [19]. Additionally, cell types like endothelial cells [35] and myeloid cells [36] are involved in the pathogenesis of pulmonary fibrosis. Even though a protective effect of NKT cells against fibrosis is observed in some studies [37], we suggest that a profibrotic polarization of the NKT cell could potentially contribute to the development of fibrosis in the lung.

### 3.2. Cell-Type-Specific Differential Genes in Support of the Profibrotic Cellular Phenotype

We identified a total of seven different cell types with sub-clusters of cells showing higher activity of the ECM pathway genes. We referred to these cells as ECM-active cell populations and considered them as representing the status of the profibrotic cellular phenotype. Next, we extracted the cell-type-specific gene signatures underpinning the profibrotic cellular phenotype by comparing the ECM-active cell populations to non-ECM-active cell populations within the same cell type (Appendix A; each sheet represents a cell type). The majority of the ECM-active cells come from the lung, which involves epithelial, endothelial, secretory, myeloid, fibroblast, and NKT cell types. Among these cell types, endothelial cells have the most uniquely up-regulated differential genes, followed by secretory cells (Figure 2A), while epithelial and endothelial cells share the most common down-regulated differential genes (Figure 2B). Up-regulated differential genes tend to be unique to each cell type, while down-regulated genes are shared among similar cell types, for example, endothelial and epithelial cells.

To characterize the functional relevance of our cell-type-specific profibrotic cellular phenotype, we conducted pathway enrichment analysis on each of the up-regulated differential gene sets first (Figure 2C). We set further requirements in the analysis, including setting the minimal pathway size (number of genes) to more than 15 in genes and setting the enriched gene percentage to more than 10%, in addition to an FDR-corrected significance level of less than 0.05. As shown in Figure 2C, the COVID-19 pathway is active across multiple cell types (adjusted *p* = 2.43 × 10^−7^ for low count fibroblast cell; *p* = 2.20 × 10^−54^ for endothelial cell; *p* = 1.02 × 10^−45^ for epithelial cell; *p* = 1.59 × 10^−58^ for fibroblast cell; *p* = 7.58 × 10^−55^ for myeloid cell; *p* = 6.21 × 10^−50^ for secretory cell; *p* = 4.08 × 10^−61^ for NKT cell), except the PCT-S3-1 in the kidney. While this result is interesting, it is expected because the scRNA-seq data was originally obtained using COVID-19 patients. Functions enriched in PCT-S3-1 cells are unique in comparison to the rest of the cell types, such as mineral absorption (adjusted *p* = 6.66 × 10^−5^), glycolysis and gluconeogenesis (adjusted *p* = 1.96 × 10^−3^), and others. Specifically, dependence on aerobic glycolysis is a feature of human cancers, including renal carcinoma, which was first described by Warburg [38]. Multiple viral infection pathways were elevated in the low-count fibroblast cells of the heart (i.e., low count fib in Figure 2C), despite the fact that no virus was detected in the heart. In particular, viral myocarditis is common during viral infections, especially COVID-19 infection [39], and is also identified as highly elevated in the profibrotic fibroblast cells in the heart (adjusted *p* = 1.38 × 10^−5^). Profibrotic cell populations of different cell types from the lung share the majority of up-regulated functional pathways. Specifically, the carcinogenesis pathway (reactive oxygen species (ROS)) is reported to mediate Epstein-Barr virus (EBV) reactivation, which plays an important role in the development of nasopharyngeal carcinoma [40]. ROS induction is part of a series of early events that may trigger chronic injury, eventually leading to the development of fibrosis [10]. Unexpectedly, we observed that genes involved in several neurodegenerative disease-related pathways were highly expressed in the profibrotic cells, which might help explain the cognitive impairments reported in COVID-19 patients after hospital discharge [41,42,43].

Pathway enrichment analysis was also conducted on the down-regulated cell-type-specific differentially expressed gene sets following the same protocol. However, few pathways were identified, among which attenuated smooth muscle contraction enriched in fibroblast cells from the lung implied the potential stiffness in pulmonary tissue in response to severe viral infection.

### 3.3. Co-Active Functional Pathways Driving the Cell-Type-Specific Profibrotic Phenotype

To gain a more comprehensive view of the elevated functional network underlying these ECM-active cell populations, we performed single-cell single-pathway enrichment analysis (SSA) using all the molecular pathways curated by the MSigDB (c2.cp collection) [26]. Based on the same SSA protocol (see Section 2 for details), we identified pathways actively expressed in the ECM-active cell population in comparison to the non-ECM-active population within each cell type, using the Fisher’s exact test. With an FDR cutoff of 0.05, we generated 49, 467, 324, 96, 218, 35, 117, and 166 significantly enriched co-active pathways, respectively, in low-count fibroblast, endothelial, epithelial, fibroblast, myeloid, PCT-S3-1, secretory, and NKT cells (Appendix A; each sheet represents a cell type).

A co-active pathway network was then constructed using the top-enriched active molecular pathways, with linkage between pathways showing the percentage of ECM-active cells shared within each cell type. Pathways like Matrisome or ECM-related pathways share more common genes with the ECM pathway. More interactions among them are expected while constructing the co-active pathway network (Figure 3). In particular, we found that the copy number variation syndrome pathway is involved in all ECM-active cell populations within the lung. As a common mechanism for developmental neuropsychiatric disorders [44], the co-activity of this pathway is actually in line with our observations regarding the multiple up-regulated neurodegenerative disease-related pathways shared by different ECM-active cell types in the lung (Figure 2C). Another shared co-active signature pathway in the lung is the biological oxidations (REACTOME Identifier: R-HSA-211859), which accounts for genomic instability and has been implicated in the pathogenesis of several human diseases like fibrosis and cancers [45].

In the heart, one of the proinflammatory signaling cascades, known as the Fc epsilon receptor (FCERI) signaling pathway, in fibroblast cells is co-active with ECM-related profibrotic pathways, which indicates a potential interaction between the profibrotic and proinflammatory cellular phenotype [46]. The FCERI pathway is also one of the top-ranked pathways shared between diabetes and COVID-19 patients [45]. Additionally, the co-active *TGF-β1* signaling pathway (KEGG; Appendix A, sheet 2 “Endothelial cell”) in endothelial cells from the lung is one of the key molecular mechanisms that leads to temporary collagen accumulation in many organs [47]. Abnormal activation of this signaling pathway will result in progressive fibrosis, which contributes to morbidity and mortality worldwide. Unexpectedly, we noticed the co-activity with the cytotoxic pathway in renal cells. This protective effect could be a result of stress response during the viral infections and might serve a protective role in potential renal injury [48].

### 3.4. Validation of Cell-Type-Specific Profibrotic Gene Signatures in Both Human and Mouse Datasets

In total, we computationally identified a profibrotic cell population spanning six different cell types in the lung, one cell type in the kidney, and one cell type in the heart. Next, we aimed to validate the ability to capture profibrotic changes in three independent single-cell datasets in both humans and mice. NES scores were calculated to indicate gene signature activity in each individual cell for the pairwise comparisons (see Section 2).

During the acute bleomycin-induced lung injury in mice, major cellular dynamics and desired responsive patterns were captured by our cell-type-specific gene signatures (Figure 4A). As demonstrated by their NES scores, myeloid, secretory, and NKT cells responded during rapid lung injury-induction as early as day 3 after being treated with bleomycin, while endothelial, epithelial, and fibroblast cells started responding to stimuli as early as day 7. Common responding patterns, regardless of cell types, peaked at day 10, which is consistent with prior observations [28]. A similar responsive trend held for NKT cells, but no significance was reached due to the insufficient and unbalanced cell numbers across the conditions.

We further checked the NES scores of our gene signatures along the natural disease courses in human lung fibrosis (Figure 4B). As expected, the discontinuity of cellular status and individual heterogeneity added another layer of complexity into the responsive patterns in humans. Compared to the rapidly induced lung fibrosis in mice, responsive patterns in the human lung with COPD and IPF were weaker. A moderate uptrend was observed in cells from control to IPF in endothelial, epithelial, and myeloid cells. A general responsive pattern on all six different cell types peaked in cells from the IPF. Unexpectedly, none of the pairwise comparisons in fibroblast cells were statistically significant. Considering the different behaviors of fibroblast cells between rapid-induced lung fibrosis in mice and naturally progressed lung fibrosis in humans, we brought in a collective single-cell dataset of fibroblasts with more cells and more data points to illustrate the dynamic contributions of its relevant gene signature along natural disease courses (Figure 4C). Collectively, this result supported that fibroblast cells may be actively involved in lungs with IPF as compared to the healthy controls. The slight downslope in its gene signature activity implied that a more complicated dynamic machinery was involved in fibroblast functionalities.

### 3.5. Utilization of Cell-Type-Specific Profibrotic Scores to Predict Survival in the TCGA Cohort

To emphasize the clinical relevance of these profibrotic cell populations, we further examined their prognosis potential in human bulk tumors from the cancer patients, using the cell-type-specific gene signatures to assign a cell-type-specific profibrotic score to each tumor sample. The cell-type-specific up- and down-regulated genes, with their average gene expression for both ECM-active and non-ECM-active phenotype within each cell type, were organized into a single cell-derived marker file. This was then used to calculate the cell-type-specific profibrotic phenotype score using bulk tumor samples from the TCGA cohort (see Section 2). A total of 2514 differential genes were included in this analysis, with 10,325 bulk tumor samples spanning 31 TCGA cancer types.

An initial check on the gene expression patterns of all the differential genes in bulk tissue showed the absence of the profibrotic signatures in the tumor-adjacent normal samples, as expected (Figure 5A, Normal panel). Cell-type-specific differential gene signatures are present in some tumor types but they are not limited to the four organ systems included in our study. For instance, differential gene signatures from fibroblast cells in the lungs from our study presented in kidney renal clear cell carcinoma (KIRC) and bladder urothelial carcinoma (BLCA) patients. Survival analysis further confirmed the prognosis value of the profibrotic cellular scores, calculated based on these cell-type-specific differential gene signatures (Figure 5B,C). Some cell types showed uniqueness to a certain type of organ system, for example, epithelial cells to the lung and PCT-S3-1 cells to the kidney. A bidirectional trend in prognosis patterns between these two cell types was also observed (Figure 5B). Profibrotic score in epithelial cells of the lung indicated a worse prognosis effect in tumor patients with a higher profibrotic score (Figure 5B, left panel, *p* = 0.023). Interestingly, the profibrotic score calculated from PCT-S3-1 cells in the kidney showed an opposite trend in kidney cancer (Figure 5B, right panel, *p* = 0.022). This observation is actually consistent with our co-active pathway analysis in the kidney; in the kidney Circos plot (Figure 3), one of the top co-active pathways imposed a protective role in potential renal lesions. In TCGA cancer types of adrenocortical carcinoma (ACC), bladder urothelial carcinoma (BLCA), liver hepatocellular carcinoma (LIHC), kidney renal clear cell carcinoma (KIRC), sarcoma (SARC), and skin cutaneous melanoma (SKCM), we found consistently worse effects of higher fibroblast-derived profibrotic scores in predicting cancer prognosis.

## 4. Discussion

Although previous studies using single-cell transcriptomics have identified some cell types and molecular pathways of pulmonary fibrosis in particular [34,49], an integrative study emphasizing common characteristics across diverse organ systems to identify early profibrotic changes at a comprehensive cell-type level are still lacking. In this study, we systematically performed single-cell single-pathway enrichment analysis and provided a single-cell landscape of profibrotic changes across multiple organ systems. In addition to fibroblast cells, we identified six previously under-recognized cell types involved in this process. Our profibrotic score derived from multiple representative organs is useful for predicting cancer prognosis.

Taking advantage of these granular transcriptomic sequencing data, we revealed a greater degree of transcriptomic heterogeneity at the cell-type level in response to viral infections. Early profibrotic changes are observed in some cell populations within a certain cell type in particular. Intriguingly, the PCT-S3 cells were split into two clusters [23]. Cluster 2 polarized into a proinflammatory phenotype, which is CFH-positive with fibrinogens (*FGB*, *FGA*) expressed. As a comparison, we report that 95.5% of the Cluster 1 in the PCT-S3 cell of the kidney shows a profibrotic phenotype, that is, it has actively expressed ECM genes. Unexpectedly, part of the NKT cell exhibits highly expressed ECM genes, which seems controversial in relation to prior research in its protective roles against fibrosis [50]. However, suppression of its antifibrogenic effects has also been demonstrated in mouse liver fibrosis [51]. Molecular pathway analysis and co-active gene set analysis further strengthen the functional relevance of the profibrotic cellular phenotype at the cell-type level to diverse human diseases, like infections, fibrosis, or even human cancer. Independent validations confirm the ability of our cell-type-specific gene signatures to capture early profibrotic changes using both human and mouse scRNA-seq datasets. Furthermore, the prognosis analysis using bulk tumor samples demonstrates the clinical relevance of our cell-type-specific differential gene signatures. Understanding the molecular mechanisms in support of the profibrotic phenotype may yield novel therapeutic targets for the early prevention of diverse human diseases, including cancer.

In addition to these notable findings, some limitations warrant discussion. First, we computationally identified profibrotic cell populations as those actively expressing ECM pathway genes. The completeness of this functional pathway may affect the performance of the identification process. Second, the high-level sparsity (dropout rates) and the large-number cells in single-cell datasets hinder our interpretations of the activities of individual molecular pathways at the single-cell level [52]. The development of more sophisticated imputing algorithms and the improvement of single-cell sequencing coverage will surely enhance the characterization of those profibrotic changes preceding a diversity of human diseases. Third, single-cell annotation is an active, but still underdeveloped, area of research. Therefore, potential biases introduced in this annotation process may affect the accuracy of the relative ranking procedure required by the adapted single-cell single-pathway enrichment analysis [27]. A workaround we applied in the pipeline is to generate relative ranks within each cell cluster to account for the similarities and biological dependence among different cell types.

Together, our results provide valuable insights into understanding the common mechanisms leading to the development of a variety of human diseases, like fibrosis or even human cancers, across multiple organ systems. This high-resolution landscape identifies novel profibrotic cell types, molecular pathways, and co-active gene sets that characterize the early profibrotic changes in response to severe viral infections. Future directions investigating and monitoring our cell-type-specific differential gene signatures in human single-cell time-series datasets should provide additional insights into the fundamental mechanisms that result in the development and establishment of fibrosis and human cancers as well as their dependent microenvironment.

## Figures and Tables

**Figure 1 cancers-13-06024-f001:**
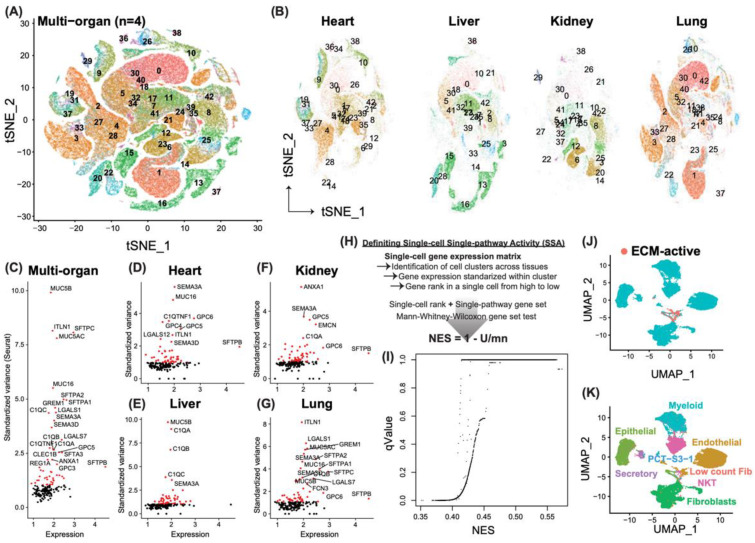
Identification of novel profibrotic cell types across multi-organ systems. (**A**) Overview of the 43 cell clusters using the single-cell tissue atlas. T-distributed stochastic neighbor embedding (t-SNE) of normalized transcriptomic data is shown. Each dot represents a single cell. (**B**) Cell clusters using t-SNE embeddings across different organ systems. (**C**) Standardized variance (defined by Seurat) of ECM pathway genes in the whole single-cell data set, with the most variable genes labeled in black. *X*-axis shows the median expression for each gene. (**D**–**G**) Standardized variance of ECM pathway genes across different organ systems, with the most variable genes labeled in black. *X*-axis shows the median expression for each gene. (**H**) Adapted computational pipeline of defining pathway activity at the single-cell level. (**I**) Dotplot showcases the distribution of single pathway activity, with normalized enrichment score (NES) plotted as *x*-axis, and significance level of q value as the *y*-axis. (**J**) Re-clustering of cells identified as ECM-active and their counterparts as non-EMC-active cells. Cells are colored based on their ECM activity. (**K**) Re-clustering of cells identified as ECM-active and their counterparts. Cell type annotations are labeled on the plot next to their corresponding clusters.

**Figure 2 cancers-13-06024-f002:**
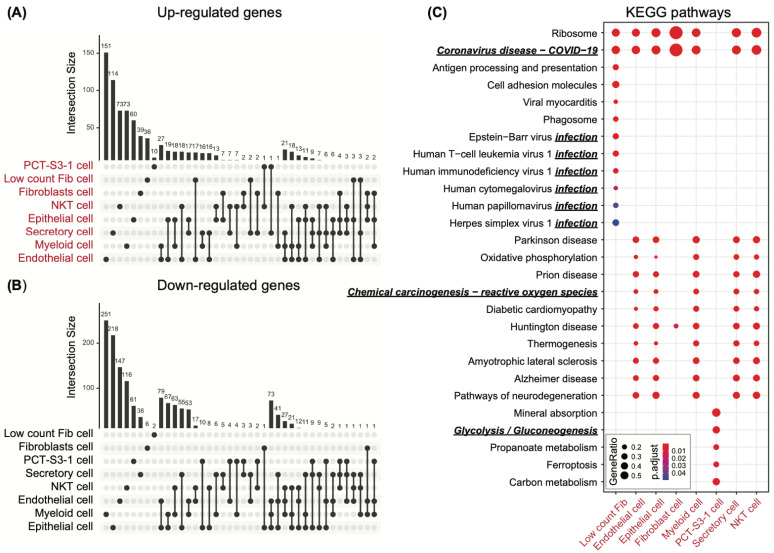
Profibrotic cell-type-specific differentially expressed genes and their KEGG pathway annotations. (**A**) Upset plot of cell-type-specific up-regulated genes while comparing ECM-active to non-ECM-active cell populations within the same cell type. (**B**) Upset plot of cell-type-specific down-regulated genes while comparing ECM-active to non-ECM-active cell populations within the same cell type. (**C**) KEGG pathway enrichment analysis using cell-type-specific up-regulated genes. Functions emphasized are labeled using italic, bold, and underlined text. Ratios for enriched genes against pathway size are filtered to be larger than 10%.

**Figure 3 cancers-13-06024-f003:**
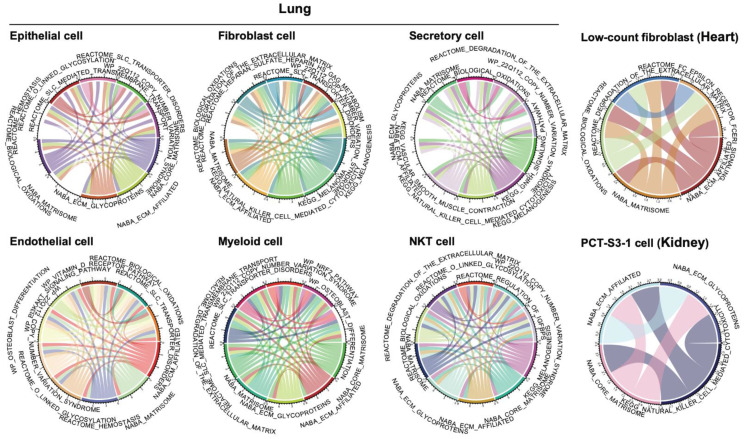
Cell-type-specific profibrotic co-active pathway network. Top-ranked co-active pathways are shown in Circos plot, with a maximum of 10 pathways illustrated. Each plot represents a cell type. Pathway labels are arched and centered to the corresponding segment in the plot. Links between different pathways represent co-activity, with their bandwidths scaled to the percentage of shared active cells.

**Figure 4 cancers-13-06024-f004:**
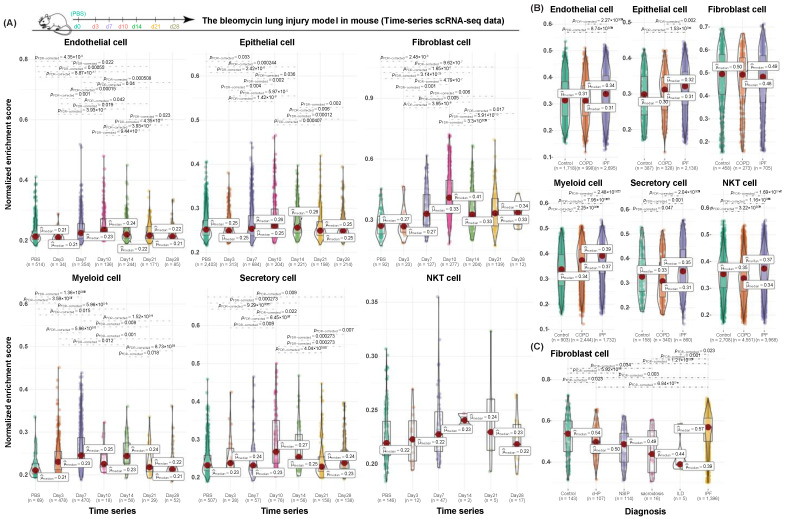
Validation of cell-type-specific gene signatures in single-cell datasets from both humans and mice. (**A**) Box-violin plot of NES (*y*-axis) on six different cell types in mouse single-cell time-series dataset. Time points are shown as *x*-axis. Illustration of longitudinal sample collections is shown on top of the whole panel. (**B**) Box-violin plot of NES (*y*-axis) on six different cell types in human lung fibrosis. (**C**) Box-violin plot of NES (*y*-axis) on fibroblast cells in human lung fibrosis. For individual box-violin plot, 25th, 50th, and 75th percentiles of the data points are shown with boxplot. Red dot represents the mean. Detailed corrected *p* values (FDR values) in the pairwise comparisons are labeled on each plot, and only the significant comparisons are shown. Number of total cells contained in each condition is included in parentheses. Abbreviations: cHP, chronic hypersensitivity pneumonitis; COPD, chronic obstructive pulmonary disease; ILD, unclassifiable chronic interstitial lung diseases; IPF, idiopathic pulmonary fibrosis; NSIP, nonspecific interstitial pneumonia.

**Figure 5 cancers-13-06024-f005:**
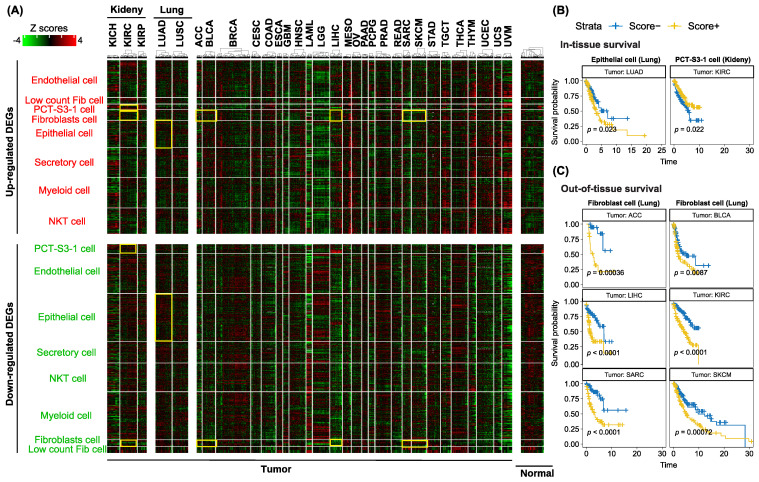
Single-cell transcriptomic profibrotic signatures in TCGA tumors and their prognosis. (**A**) Heatmap showing cell-type-specific differential gene signatures across all the TCGA tumor and adjacent normal samples. Each row is a differential gene, while each column is a tumor or normal sample. Gene expression is row-scaled and capped at +/− 4 standard deviations. Heatmap color scheme is shown on the top left corner, with green representing low gene expression and red indicating high gene expression. Rows and columns are aggregated into categories of cell types, and different tumor types and adjacent normal groups, respectively. (**B**) The Kaplan-Meier survival curve, using epithelial and PCT-S3-1 cell-derived profibrotic scores to predict prognosis in tumor patients of LUAD and KIRC, separately. (**C**) The Kaplan-Meier survival curve, using fibroblast cell-derived profibrotic scores to predict prognosis in patients of TCGA tumor type of ACC, BLCA, LIHC, KIRC, SARC, and SKCM. Fibroblast cell is originally native to the lung.

## Data Availability

Multi-organ tissue atlas at single-cell resolution is available under accession number GSE171668 at Gene Expression Omnibus (GEO, https://www.ncbi.nlm.nih.gov/geo/, accessed 6 July 2021). Mouse time-series scRNA-seq data of the bleomycin-induced lung injury is available under accession number GSE141259 at GEO. Human single-cell atlas of idiopathic pulmonary fibrosis is available under accession number GSE136831 at GEO. A collection of human fibroblast cells spanning different disease courses in lung fibrosis is available under accession number GSE135893 at GEO. TCGA gene expression data and clinical outcome information are available at The Genomic Data Commons (GDC; https://gdc.cancer.gov/about-data/publications; accessed 6 July 2021).

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
