# Peer review of "Cell-Type-Specific Profibrotic Scores across Multi-Organ Systems Predict Cancer Prognosis"

_cancers, 2021, doi:10.3390/cancers13236024_

Round 1

Reviewer 1 Report

The main aim of the paper  was aimed to computationally identify profibrotic cell populations using single-cell transcriptomic data and to identify gene signatures that could predict cancer prognosis.

I think this is an interesting and well-organized paper.

The topic is quite original.

The paper is well written and the text is clear and easy to read.

The conclusions are consistent with the evidence and arguments presented, and they address the main question posed.

Use profibrotic scores in order to predict cancer prognosis  is the main question addressed by the research. As single-cell transcriptomic data across multi-organ systems was used in order to predict cancer prognosis in patients using profibrotic scores. Compared with other published material, it add the prediction of cancer prognosis. The conclusions are consistent with tha arguments presented and they address the main question posed. The references appropriate and figures are interesting.

Author Response

Attached please find the response in word file.

Reviewer 2 Report

The authors follow a conceptually interesting idea that has a lot of merit: can you detect pro-fibrotic cell types across tissues through deployment of single cell RNA-sequencing. The difficulty in this idea is that fibrotic tissues are a combination of cell types and thus it seems likey that profibrotic lesions are a combination of different transitional states rather than one specific cell type which is easily capturable by scRNA-seq.

The authors address this nicely in the results 3.2 section where they define profibrotic cell type specific gene signatures. 

The biggest weakness of this paper is that there is no validation of their signatures in a different set of conditions. The leveraging of TCGA data in results section 3.4 does demonstrate clinical relevance, however it is not clear that these gene cluster signatures are clinically relevant as the Kaplan Myer Curves are based on predicted prognosis vs. actual prognosis. A validation of the signature either by actual patient outcome data from TCGA or validation by a separate set of sc-RNA seq in cases of fibrosis would better suffice to serve the authors points. 
